# The Impact of a Web-Based Lifestyle Educational Program (‘Living Better’) Reintervention on Hypertensive Overweight or Obese Patients

**DOI:** 10.3390/nu14112235

**Published:** 2022-05-27

**Authors:** Pedro Múzquiz-Barberá, Marta Ruiz-Cortés, Rocío Herrero, María Dolores Vara, Tamara Escrivá-Martínez, Raquel Carcelén, Rosa María Baños, Enrique Rodilla, Juan Francisco Lisón

**Affiliations:** 1Department of Nursing and Physiotherapy, Faculty of Health Sciences, University CEU-Cardenal Herrera, CEU Universities, 46115 Valencia, Spain; pedro.muzquizbarbera@uchceu.es; 2Department of Biomedical Sciences, Faculty of Health Sciences, University CEU-Cardenal Herrera, CEU Universities, 46115 Valencia, Spain; marta.ruiz3@alumnos.uchceu.es (M.R.-C.); juanfran@uchceu.es (J.F.L.); 3Department of Psychology and Sociology, Universidad de Zaragoza, 50009 Teruel, Spain; rocio.herrero@uv.es; 4Centre of Physiopathology of Obesity and Nutrition (CIBERobn), CB06/03/0052, Instituto de Salud Carlos III, 46115 Valencia, Spain; m.dolores.vara@uv.es (M.D.V.); tamara.escriva@uv.es (T.E.-M.); rosa.banos@uv.es (R.M.B.); 5Polibienestar Research Institute, Universitat de València, 46022 Valencia, Spain; 6Department of Medicine and Surgery, Faculty of Health Sciences, University CEU-Cardenal Herrera, CEU Universities, 46115 Valencia, Spain; raquel.carcelen@uchceu.es; 7Hypertension and Vascular Risk Unit, Hospital Universitario de Sagunto, 46520 Valencia, Spain

**Keywords:** internet, eHealth, lifestyle, Mediterranean diet, physical activity, weight loss, obesity, hypertension

## Abstract

‘Living Better’, a self-administered web-based intervention, designed to facilitate lifestyle changes, has already shown positive short- and medium-term health benefits in patients with an obesity–hypertension phenotype. The objectives of this study were: (1) to examine the long-term (3-year) evolution of a group of hypertensive overweight or obese patients who had already followed the ‘Living Better’ program; (2) to analyze the effects of completing this program a second time (reintervention) during the COVID-19 pandemic. A quasi-experimental design was used. We recruited 29 individuals from the 105 who had participated in our first study. We assessed and compared their systolic and diastolic blood pressure (SBP and DBP), body mass index (BMI), eating behavior, and physical activity (PA) level (reported as METs-min/week), at Time 0 (first intervention follow-up), Time 1 (before the reintervention), and Time 2 (post-reintervention). Our results showed significant improvements between Time 1 and Time 2 in SBP (−4.7 (−8.7 to −0.7); *p* = 0.017), DBP (−3.5 (−6.2 to −0.8); *p* = 0.009), BMI (−0.7 (−1.0 to −0.4); *p* < 0.001), emotional eating (−2.8 (−5.1 to −0.5); *p* = 0.012), external eating (−1.1 (−2.1 to −0.1); *p* = 0.039), and PA (Time 1: 2308 ± 2266; Time 2: 3203 ± 3314; *p* = 0.030, Z = −2.17). Statistical analysis showed no significant differences in SPB, DBP, BMI, and eating behavior between Time 0 and Time 1 (*p* > 0.24). Implementation of the ‘Living Better’ program maintained positive long-term (3-year) health benefits in patients with an obesity–hypertension phenotype. Moreover, a reintervention with this program during the COVID-19 pandemic produced significant improvements in blood pressure, BMI, eating behavior, and PA.

## 1. Introduction

The implementation of strategies that effectively promote the prevention and treatment of the obesity–hypertension phenotype is urgently required. These must have the clinical objectives of controlling blood pressure (BP) and body composition (fat loss and muscle mass gain), improving cardiorespiratory fitness and functional capacity, and reducing polypharmacy [1]. In this sense, the most recent clinical guidelines on hypertension (HTN) and obesity (OB) [2,3,4] agree that promoting a healthy lifestyle should be the first step considered in obese patients with HTN. To achieve these changes, the process must be based on two fundamental pillars: regular physical activity (PA) and healthy eating behavior [5].

PA has been defined as “any bodily movement produced by skeletal muscles that results in energy expenditure” [6]. Specifically, exercise is described as “a subset of physical activity that is planned, structured, and repetitive and has as a final or an immediate objective the improvement of maintenance of physical fitness” [6]. At present, exercise is considered a polypill for the prevention and treatment of numerous health conditions, including chronic diseases such as OB and HTN [5]. Thus, it has been shown that regularly engaging in sustained PA over time is essential to maintain long-term weight loss [1,7]. However, even though moderate weight reductions (~1–3 kg) can be achieved with exercise programs without dietary modifications, the combination of regular PA and healthy eating behaviors, including a decrease in caloric intake [8], is the most effective strategy to address weight loss and its maintenance [1]. Of note, one of the most successful dietary interventions described in the academic literature was from the Prevención con Dieta Mediterránea (PREDIMED) study, and was precisely administered in a Spanish population at high risk for cardiovascular events [9].

The recent academic literature indicates that there was a dramatic decrease in PA during the COVID-19 pandemic, which was especially worrisome in patients with associated metabolic conditions [10]. In hypertensive older adults, unhealthy changes manifested as a reduction in PA and increased sedentary behavior [11]. Other research suggested that unhealthy eating patterns intensified among high-risk patient groups during the pandemic [12]. Similarly, a related study showed that obese individuals spent less time engaging in PA, exercised less intensely, and were more anxious about eating during the pandemic, all of which can make body weight control more difficult [13].

In this context, online interventions can reach different populations, overcoming barriers and limitations, and they represent effective strategies for the prevention and/or treatment of multiple health conditions. However, to date, only four studies [14,15,16,17] have analyzed the effectiveness of such treatments in patients with both health conditions—that is, in individuals presenting an OB-HTN phenotype. Of note, none of the four studies performed in this specific area, and only one in non-hypertensive obese adults [18], followed up with patients who had completed an online educational intervention for at least 3 years. In addition, to the best of our knowledge, no research has yet analyzed the effects of a second intervention (reintervention) with an online intervention program in patients with OB, HTN, or any other type of cardiovascular disease.

Given all the above, in this current study, we set out to (1) understand the evolution at 3 years of a group of hypertensive overweight or obese individuals who had followed the ‘Living Better’ web-based program in 2018 [17]; (2) analyze the effects of completing this program a second time (reintervention) during the COVID-19 pandemic—3 years after the initial intervention—in terms of systolic and diastolic blood pressure (SBP and DBP), body mass index (BMI), number of antihypertensive drugs used, PA, eating behavior, and adherence to the Mediterranean diet.

## 2. Materials and Methods

### 2.1. Study Design

This was a prospective quasi-experimental study (ClinicalTrials.gov: NCT04571450). This research was approved by the Human Research Ethics Committee at the Hospital Universitario de Sagunto and followed the ethical guidelines established in the Declaration of Helsinki. The study describes the 3-year follow-up of 29 patients who already received an intervention with the ‘Living Better’ program in 2018 (*n* = 105) [17], as well as the result of completing the same program for a second time (reintervention) with the aim of helping to minimize the negative impact of the COVID-19 pandemic on lifestyle. To analyze the long-term effects of the program, we used the follow-up values obtained at the end of the first study as our starting point (Time 0) [17], which was carried out 9 months after this intervention. We also used the different variables analyzed in 2018 and recorded shortly before the start of the second intervention (Time 1), 21 months later than Time 0. Once this evaluation was completed, the participants started the 3-month online reintervention with the ‘Living Better’ program. Finally, in order to understand the impact of this second intervention on the health of the participants, all the variables were recorded again at the end of the program (Time 2) (Figure 1). Of note, all the patients were assessed within 3 days at the different study timepoints.

### 2.2. Eligibility Criteria 

Because this was a reintervention study, the first inclusion criterion was that the patients had participated in the ‘Living Better’ online program in 2018. In addition, we used the same inclusion and exclusion criteria that we applied in the first study [17]: overweight (BMI between 24.9 kg/m^2^ and 30 kg/m^2^) or type I obese (29.9 kg/m^2^ < BMI < 35 kg/m^2^) adults aged 18 to 65 years with HTN. HTN was defined as SBP ≥ 140 mmHg or DBP ≥ 90 mmHg, or patients who take antihypertensive drugs. We also used the same exclusion criteria: a diagnosis of diabetes, previous ischemic heart or cerebrovascular disease, serious psychiatric disorders, use of more than three antihypertensive medicines, physical impairments that could hinder engagement in PA, receiving other treatments for weight loss, or no access to the Internet.

### 2.3. Procedure

This study was carried out in the Hypertension and Vascular Risk Unit at the Hospital Universitario de Sagunto (Valencia, Spain) from January 2018 to January 2021 (reintervention period from October 2020 to January 2021). We used the hospital postal service to contact the 105 participants who took part in the 2018 study. Of those who agreed to participate again, the final sample in this work comprised a total of 29 participants (Figure 2). After obtaining their informed consent for participation in the study and formalizing their registration, we incorporated the participants into a single experimental group, which received a 3-month reintervention via the ‘Living Better’ web-based platform. Furthermore, we telephoned all these individuals to remind them of the program details and to resolve any questions they had.

### 2.4. Intervention

We used the same ‘Living Better’ program, implemented via the Internet, as in the 2018 study [16,17]. This multimedia, interactive, and self-administered program comprises 9 intervention modules that try to gradually change the eating behavior and PA patterns of the participants. All the modules include videos, texts, tasks, daily records, and files that the patient can download to work on the content. Considering the suggestions of the participants after the first intervention, on this occasion, we had converted part of the written content into audiovisual materials to help facilitate usability. However, despite these changes, the program content was identical to that of the first intervention. More details about the first intervention can be found in Mensorio et al. [16], Lisón et al. [17], Baños et al. [19], and also in the Appendix A (Appendix A and Video S1).

### 2.5. Outcome Measures

#### 2.5.1. Systolic and Diastolic Blood Pressure

To avoid coronavirus infections, the health authorities and hospital regulations prohibited access to medical facilities for patients who did not need urgent healthcare. Therefore, the participants were asked to visit a pharmacy close to their home so that the same person (a pharmacist or pharmacy assistant) could record these variables at Time 1 and Time 2 using the same approved device. As in the first intervention, the participants were instructed to record measurements between 8 a.m. and 12:00 p.m. noon to minimize variability in their daytime BP figures. BP was strictly analyzed according to the European Society of Hypertension (ESH)/European Society of Cardiology guidelines and the American College of Cardiology/American Society of Hypertension [2,3], so measurements were performed in the sitting position, in a quiet environment for 5 min before BP measurements, avoiding prior consumption of alcohol or smoking, drinking caffeine, or engaging in strenuous exercise. Three BP measurements were recorded, 1 minute apart, and BP was calculated as the average of the last two BP readings. Additional measurements were performed if the first two readings differed by more than 10 mmHg. Of note, the participants of this study—as with, in general, every patient treated in the Hypertension Unit at the Hospital Universitario de Sagunto—were routinely trained to correctly measure BP in this way.

#### 2.5.2. Weight, Height, and BMI 

Because of the aforementioned COVID-19-related health concerns, these variables were also recorded in local pharmacies, following the same indications. Specifically, clothing was standardized during weight measurement, and patients were instructed to visit the pharmacy while fasting and preferably always at the same time to avoid the possibility that any food or drink ingested could influence their data. BMI was calculated by dividing patient weight by their height squared (kg/m^2^).

#### 2.5.3. Antihypertensive Drugs

The patient registered the number of antihypertensive drugs they used through the intervention program platform.

#### 2.5.4. Physical Activity Levels

The short version of the International Physical Activity Questionnaire (IPAQ-SF) was used [20,21] to assess the time that each subject had spent being active in the 7 days prior to completion of the survey. Different scores are awarded in the IPAQ-Short, depending on the time spent engaging in moderate or vigorous activities, walking, or sitting each week. The unit of measurement for this questionnaire is METs-min/week, which expresses the average of each individual’s metabolic expenditure per minute while engaging in weekly PA. Thus, higher figures reflect a higher level of activity, while lower values express a lower level of weekly PA [20,21]. Data should be interpreted using the formula published by Ainsworth et al. [22] to classify their PA levels as high ( >1500 METs-min/week), moderate (600–1500 METs-min/week), or low (<600 METs-min/week).

#### 2.5.5. Eating Behavior

To analyze the eating behavior of the patients, we employed the ‘Dutch Eating Behavior Questionnaire’ (DEBQ) [23,24], which comprises 33 items and uses a 5-point Likert scale to evaluate 3 eating styles, emotional eating (13 items), external eating (10 items), and restrained eating (10 items), with higher scores indicating greater agreement with the eating behavior statements.

#### 2.5.6. Adherence to the Mediterranean Diet

Eating habits were recorded before (Time 1) and after (Time 2) the reintervention using the ‘Mediterranean Diet Adherence Screener’ (MEDAS) from the PREDIMED study [25]. This questionnaire assesses adherence to the Mediterranean diet through 14 items, 12 of which are related to the frequency of food consumption, while 2 are about dietary habits linked to the Mediterranean diet. Each item is scored with a value of 0 or 1 and, based on the final score, the patients were classified as having low (0–5 points), medium (6–9 points), or high (≥10 points) adherence to the Mediterranean diet.

#### 2.5.7. Satisfaction with the Reintervention

As in the first study [17], this variable was evaluated on a scale from 0 (minimum satisfaction) to 10 (maximum satisfaction).

#### 2.5.8. Adherence to Reintervention

This was analyzed through the data registered by the participants on the platform. This also allowed us to gauge the degree of completion of the different program modules by each participant—in other words, how many of the nine modules they had reviewed. 

### 2.6. Statistical Analysis

The statistical analyses were performed according to the intention-to-treat paradigm using SPSS software (version 19.0; IBM Corp., Armonk, NY, USA) for Windows, and the statistical significance was set at *p* < 0.05 for all our analyses. The data in this study are presented as mean (SD). Compliance with the assumption of normality was checked for each dependent variable using the Shapiro–Wilk test. One-way ANOVA tests followed by Bonferroni post-hoc tests were performed for the variables that met the assumption of normality (SBP, DBP, BMI, and eating behavior). The effect sizes were estimated using the ηp^2^ and were interpreted following Cohen’s guidelines for small, moderate, and large effect sizes (ηp^2^ = 0.01, 0.06, or 0.14, respectively). Friedman tests followed by non-parametric Wilcoxon tests to compare the three study timepoints (Time 0, Time 1, and Time 2) were used for the variables that violated the assumption of normality (PA and antihypertensive drugs). In addition, *t*-tests for related samples were performed to compare the level of adherence to the Mediterranean diet before and after the reintervention (Time 1 vs. Time 2), as well as to contrast the degree of participant satisfaction after the reintervention compared with the first intervention. Adherence to the reintervention was estimated by calculating the average percentage of the 9 ‘Living Better’ program modules completed by the 29 participants. Finally, at Time 0, depending on whether the assumption of normality was fulfilled, *t*-tests (for independent samples) or Mann–Whitney U tests were carried out for the different study variables to compare the 29 reintervention participants to the 76 participants excluded from this study.

## 3. Results

### 3.1. Reported Changes in the SBP, DBP, BMI, and Eating Behavior

Table 1 shows the patient values for the variables prior to the second intervention (Time 1). Specifically, regarding BMI, 62% of the participants (18 of 29) were overweight at Time 1, while the other 11 patients (38%) had type I obesity. In addition, Table 2 shows data reported from the different timepoints and the results of the post-hoc ANOVA analysis. As shown, there were significant differences between the start of the second intervention (Time 1) and the end of the program (Time 2) in all variables—except for restrained eating—with statistically significant improvements and large effect sizes (ηp^2^ > 0.21) after completing the reintervention. However, the statistical analysis did not show significant differences between the end of the first intervention (Time 0) and the beginning of the second one (Time 1) for any of these four variables (*p* > 0.24).

### 3.2. Differences Found in Antihypertensive Drugs and PA

All patients in our study were receiving antihypertensive treatment. The Friedman test did not indicate any statistically significant changes in the number of antihypertensive drugs used between the different evaluation points (Time 0: 1.7 ± 1.2; Time 1: 1.6 ± 1.4; and Time 2: 1.6 ± 1.4; *p* = 0.439), although there were significant differences for PA (Time 0: 4024 ± 3676; Time 1: 2308 ± 2266; and Time 2: 3203 ± 3314; *p* = 0.005). Specifically, the results of the Wilcoxon tests showed differences between Times 0 and 1 (*p* = 0.015, Z = −2.43) and Times 1 and 2 (*p* = 0.030, Z = −2.17). 

### 3.3. Results Analyses of Adherence to the Mediterranean Diet, Satisfaction, and Adherence to the Reintervention

Regarding adherence to the Mediterranean diet, *t*-tests showed that there were no statistically significant differences (*p* = 0.100) between the time before (Time 1: 8.2 ± 2.1) and immediately after the reintervention (Time 2: 8.8 ± 1.7). Furthermore, participants reported a higher level of satisfaction with the program after the second intervention compared to the first one, although these findings did not reach statistical significance (first intervention: 6.8 ± 2.3, second intervention: 8.0 ± 1.4; *p* = 0.080). With regard to adherence to the reintervention, seven patients withdrew before completing the first module, 66% of the 29 participants had looked at more than half of the program (at least 5 of the 9 modules), and 38% had completed all of it. Finally, at Time 0, the comparison between the 29 volunteers who agreed to participate in the reintervention and the 76 participants excluded from the study showed no statistically significant differences in any of the studied variables (*p* > 0.29), except for the BMI, which was higher in the excluded patient group (29.0 ± 2.5 and 30.2 ± 2.8, respectively; *p* = 0.033). However, a subsequent analysis verified that the differences had already existed before the first intervention between these two groups (29.3 ± 2.6 versus 30.5 ± 2.6; *p* = 0.033). 

## 4. Discussion

This study indicates that the 29 hypertensive overweight or obese patients enrolled in the reintervention had maintained long-term benefits in terms of reduced BMI and BP at a 3-year follow-up after having completed the ‘Living Better’ online intervention [16,17]. Likewise, our results show that these variables significantly improved after the same group of patients repeated the program a second time (reintervention). To the best of our knowledge, this is the first work using a web-based program aimed at promoting a healthy lifestyle based on psychoeducation, regular engagement in PA, and the establishment of healthy eating behavior with such a long-term follow-up time. It is also the first study to describe the effects of a reintervention in patients with an OB-HTN phenotype. 

Our results did not show any significant changes in any of the study variables (SBP, DBP, BMI, antihypertensive drugs, or eating behavior) at the 3-year follow-up, compared to the first intervention in 2018, with the exception of the level of PA, which had significantly worsened. This decline may have been because of the restrictions to movements and access to sports spaces imposed by governmental authorities as a result of the COVID-19 pandemic at the time of this work. In this sense, recent research indicates that there was a significant decrease in PA at this time, accompanied by an increase in sedentary habits, due to these restrictions [10,11]. Also of note, the eating behavior of the study patients did not significantly worsen during that time. Indeed, the ‘Living Better’ program has already been shown to effectively improve emotional eating and other psychological variables related to eating and quality of life (anxiety and stress) [16]. These results are consistent with the absence of significant changes in BP and BMI, together indicating the long-term effectiveness of the ‘Living Better’ program.

To help deal with the possible negative lifestyle effects of the COVID-19 pandemic on patients with the OB-HTN phenotype (for example, decreased PA), we decided to implement a second intervention with the same program. Given the self-administered, interactive, multimedia, and web-based nature of the platform, we hypothesized that repeating this program could reinforce and enhance the knowledge that the patients had acquired after the first intervention, helping them to face the barriers and thereby perhaps minimizing the negative impact of the situation on their lifestyle and health. 

The results that we obtained after administering the reintervention confirmed our hypothesis. Thus, despite the restrictions imposed by the pandemic, the participants had significantly increased their levels of PA—after 3 months of reintervention—by approximately 30%, or around 900 METs-min/week. In addition to the improvements in PA, as already demonstrated in the first intervention in 2018 [16], reintervention with the ‘Living Better’ program also positively influenced emotional eating and external eating. In fact, one of the goals of this program is to change eating behavior (generating a more conscious and less impulsive eating style) by using psychoeducation, eating tricks, and self-control strategies. This finding is relevant because eating styles are considered to be multi-dimensional, stable, and related to OB [26]. The latter is important in the context of the negative emotions such as anxiety and panic generated by the COVID-19 pandemic, which have been associated with unhealthy eating behavior in populations with higher rates of OB [27,28]. Furthermore, adherence to the Mediterranean diet before reintervention was close to the upper limit of the ‘medium adherence’ range (8.2 points on the MEDAS questionnaire) [25], perhaps because of the effect of the first intervention. Nonetheless, the reintervention still produced a slight increase in the score by 0.6 points.

Therefore, presumably as a consequence of improvements in PA and eating behavior after the reintervention, the participants had reduced their body weight by an average of 2 kg, which translated into a significant reduction in BMI by 0.7 kg/m^2^. Of special note, this BMI reduction was even higher than that achieved after the first intervention in 2018 (0.4 kg/m^2^) [17]. In addition, the literature also reflects the direct impact that weight loss has on BP values [29]. In this sense, compared to our first study [17], the SBP and DBP of the reintervened patients also decreased further, possibly as a consequence of the greater BMI reduction. In these patients, SBP and DBP decreased by 4.7 and 3.5 mmHg, respectively (*p* = 0.017 and *p* = 0.009), compared to the non-significant reduction in SBP (−2.6 mmHg, *p* = 0.15) and the lower reduction in DBP (−2.2 mmHg, *p* = 0.05) that we reported after the first intervention in 2018. These post-reintervention improvements also exceeded those reported in the meta-analysis by Liu et al. on Internet-based lifestyle counselling [30], in which SBP and DBP were reduced by a mean of 3.8 mmHg and 2.1 mmHg, respectively. Likewise, it is important to note that the improvements that we found in this research were not the result of a change in medication, because no significant differences were reported by the participants at any of the timepoints examined in the number of antihypertensive drugs used.

In terms of program engagement [31], the percentage of participants who completed our entire program was lower (38%) than in our first intervention [17] or similar e-counselling lifestyle interventions [32]. The low completion rate for the whole program during the reintervention may have been partly because of the limitations caused by the COVID-19 pandemic, perhaps forcing the population to adapt their working hours and spaces, as well as reducing the availability of personal time and resources [33,34]. This phenomenon may also have been because the participants had remembered some of the educational content from the first intervention, leading them to complete only the modules that they considered necessary. Indeed, two thirds of the participants completed at least half of the ‘Living Better’ program (five or more modules). Moreover, the mean participant satisfaction with the reintervention was 1.2 points (out of 10) higher than the average from the first study [17], although this did not reach statistical significance. This difference may be because of the alterations we made to the program presentation by including more audiovisual content [35,36], as suggested by the patients after the first intervention.

At this point, it is important to highlight that the Internet has been shown as an effective means to promote healthy lifestyles in order to help prevent and treat chronic diseases. This is because it can reach more people (including those with limited access to health services or low levels of social support) and it can provide patients with more intensive contact with clinicians at a lower economic cost than conventional face-to-face programs [37,38]. Additionally, Internet-based platforms can provide immediate, easily accessible, individually tailored (one-on-one), and permanent (accessible at any time) support to patients in the comfort of their own homes. All these advantages were especially relevant in the context of the COVID-19 pandemic, which was ongoing while this study was implemented. Therefore, the long-term effects of the web-based ‘Living Better’ program and those obtained after a reintervention with the same program were remarkable and should be scientifically valued. They minimized the profound negative impact of COVID-19 on the health of these patients—who all had an OB-HTN phenotype—and even managed to improve their health profiles.

### Limitations

The main limitation of this study was the absence of a control group. Of note, since the sample size was small—because the study design was a continuation of previous work—and mostly for ethical reasons, all 29 participants were assigned to a single experimental group so that this population, which was especially vulnerable to COVID-19, received effective treatment during this trial. Although the positive eating behavior, PA, BMI, and BP results were similar to those obtained in our previous ‘Living Better’ randomized controlled trial, the absence of a control group must be considered when interpreting the effects of this reintervention. In addition, although, prior to reintervention, we were unable to identify any differences in the variables in the 29 participants and the 76 patients excluded from the study, we cannot rule out the possibility of a selection bias. Therefore, these findings should be interpreted with caution. Finally, the participants were unable to go to the hospital for BMI and BP measurements before and after the reintervention because of the COVID-19 pandemic restrictions. However, this problem was mitigated by having these measurements completed by the same person (a pharmacist or pharmacy assistant) using the same approved devices both times, and strictly following the ESH protocol, as in the first intervention. 

## 5. Conclusions

This study shows that the ‘Living Better’ web-based program had long-term (3-year) benefits for the health of patients with an obesity–hypertension phenotype. In addition, given the context of the COVID-19 pandemic, we evaluated the effects of implementing a second intervention in these patients with the same program to try to reduce the potential negative consequences on their lifestyles. The reintervention showed significant improvements, for the second time, in eating behavior, physical activity levels, BMI, and blood pressure.

## Figures and Tables

**Figure 1 nutrients-14-02235-f001:**
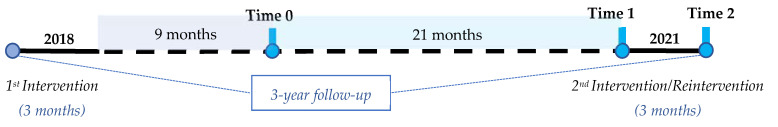
Measurements at trial profile (Time 0, Time 1, and Time 2). Figure 1 shows the time periods when assessments and reintervention were carried out, along the 3-year follow-up.

**Figure 2 nutrients-14-02235-f002:**
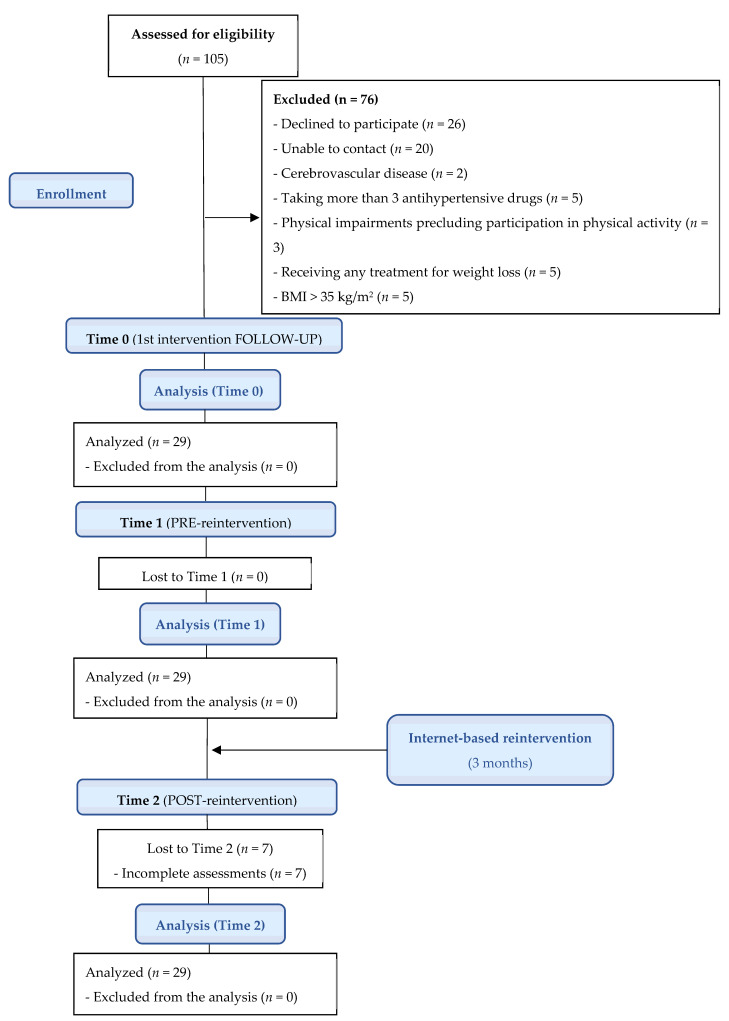
Progression of the participants through the study.

**Table 1 nutrients-14-02235-t001:** Participant characteristics.

VARIABLES		Time 1; Mean (SD) ^a^
Sex (*n*)	Women	8
	Men	21
Age (years)		57.3 (10.0)
Systolic blood pressure (mmHg)		129.6 (12.2)
Diastolic blood pressure (mmHg)		78.6 (8.1)
Weight (kg)		84.1 (11.0)
BMI (kg/m^2^)		29.2 (2.4)
Antihypertensive drugs (*n*)		1.6 (1.4)
Physical activity level (METs-min/week)		2308 (2266)
Eating behavior (points)	Emotional eating	27.1 (10.7)
	External eating	28.4 (6.6)
	Restrained eating	27.0 (6.0)
Adherence to the Mediterranean diet (points)		8.2 (2.1)

^a^ Time 1 (average values prior to patient reintervention).

**Table 2 nutrients-14-02235-t002:** Comparisons for Time 0 versus Time 1 versus Time 2.

						Time 0 vs. Time 1		Time 1 vs. Time 2	
VARIABLES		Baseline ^a^	Time 0 ^b^	Time 1 ^c^	Time 2 ^d^	Difference ^e^ (95% CI)	*p*	Difference ^f^ (95% CI)	*p*
Systolic blood pressure (mmHg)		128.8 (11.5)	127.3 (12.7)	129.6 (12.2)	124.9 (11.1)	2.3 (−4.0 to 8.5)	1.000	−4.7 (−8.7 to −0.7)	0.017 *
Diastolic blood pressure (mmHg)		77.0 (6.6)	76.4 (6.7)	78.6 (8.1)	75.1 (8.9)	2.2 (−2.0 to 6.4)	0.600	−3.5 (−6.2 to −0.8)	0.009 **
BMI (kg/m^2^)		29.3 (2.6)	28.9 (2.5)	29.2 (2.4)	28.6 (2.3)	0.3 (−0.4 to 1.0)	0.895	−0.7 (−1.0 to −0.4)	<0.001 **
Eating behavior (points)	Emotional eating	28.8 (10.6)	27.8 (8.6)	27.1 (10.7)	24.3 (9.0)	−0.8 (−3.7 to 2.2)	1.000	−2.8 (−5.1 to −0.5)	0.012 *
	External eating	30.6 (6.1)	29.5 (6.4)	28.4 (6.6)	27.3 (7.0)	−1.1 (−3.3 to 1.1)	0.640	−1.1 (−2.1 to −0.1)	0.039 *
	Restrained eating	27.9 (6.6)	28.6 (6.6)	27.0 (6.0)	26.9 (6.0)	−1.6 (−3.9 to 0.7)	0.248	−0.2 (−1.6 to 1.2)	1.000

^a^ Baseline: average values obtained prior to the first intervention, presented as mean (SD). ^b^ Time 0: average values obtained at the end of the first study, presented as mean (SD). ^c^ Time 1: average values prior to patient reintervention, presented as mean (SD). ^d^ Time 2: average values post patient reintervention, presented as mean (SD). ^e^ Difference was calculated as Time 1 (PRE-reintervention) minus Time 0 (1st intervention FOLLOW-UP). ^f^ Difference was calculated as Time 2 (POST-reintervention) minus Time 1 (PRE-reintervention). * *p* ≤ 0.05; ** *p* ≤ 0.01.

## Data Availability

The data that support the findings of this study are available from the corresponding author upon reasonable request.

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
