# Peer review of "The Impact of a Web-Based Lifestyle Educational Program (‘Living Better’) Reintervention on Hypertensive Overweight or Obese Patients"

_nutrients, 2022, doi:10.3390/nu14112235_

Round 1

Reviewer 1 Report

1. Fig. 2. Lost to time 2 n=7, but there was no change in patient numbers at time 1 and time 2?
2. Please provide data about BP, BMI and eating behaviours for all time points.

Author Response

We really appreciate your valuable comments and your effort in reviewing our paper. Thank you very much. We believe that with your comments/suggestions the paper has improved. Please see below the answers.

Point 1: Fig. 2. Lost to time 2 n=7, but there was no change in patient numbers at time 1 and time 2?

Response 1: Thank you for the question. Effectively, as shown in the Figure 2 (flowchart), 7 participants were lost to Time 2 since they did not complete the assessments at that point, but they were included in the statistical analyses according to the intention-to-treat paradigm. It has been clarified in the methods section (subsection 2.3.) of the manuscript, adding some information in the flowchart, as follows: “Uncompleted assessments (n=7)”

Point 2: Please provide data about BP, BMI and eating behaviours for all time points.

Response 2: Thank you very much for the comment, we certainly agree. As a result, we have provided all this data in the results section (subsection 3.1. Table 2), reporting the measurements obtained for SBP, DBP, BMI and eating behavior at the different study timepoints (Time 0, Time 1, and Time 2).

Reviewer 2 Report

Dear authors, congratulations on your interesting manuscript. The introduction is consistent and describes the most important issues that introduce the reader to the topic. The material and methods are fully described. Study limitations also described in detail. I only have a few small suggestions.
1. I suggest that 9 modules (module names) be presented graphically in the form of a figure.
2. Subsection 3.1- Please add in the text what percentage of patients was overweight and what percentage was obesity of the 1st degree.
  3. Subsection 3.2 - Were 100% of patients taking medication for hypertension? What percentage of the 29 patients were taking such drugs?

Reviewer 3 Report

Thank you authors for submitting your manuscript assessing the long-term effects of an online life style intervention , as well as the efficacy of repeat administration of this intervention in populations with hypertension and obesity. 

As noted, the population pool is a convenience sample and from the originally enrolled 105, 29 were agreeing and eligible to undergo the same intervention a second time. Therefore, all findings must be interpreted with caution.

It is impressive to see this kind of study being executed during a COVID surge in Spain, and the reviewer would like to congratulate the research team on pulling this off.

In general, the report will benefit from a revised figure for clarity, as well as a clear distinction between exercise and physical activity in your introduction.

Please find additional comments below:

Line 25: What is meant by quasi-experimental uncontrolled design?

Line 28: What is the timing for first intervention “follow-up”? Later it is reported there is no statistical difference between Time 0 and 1 for most of the primary outcomes listed. Are the time points Time 0, Time 1 and Time 2 evenly spaced across participants?

Line 32: What physical activity metric is this z-score for? Total PA counts? Light activity? Moderate to High Intensity? Please clarify.

Line 51-52: Please define “exercise”, as it is different from “physical activity”. Caspersen CJ, Powell KE, Christenson GM. Physical activity, exercise, and physical fitness: definitions and distinctions for health-related research. Public Health Rep. 1985;100(2):126-131 “Physical activity is defined as any bodily movement produced by skeletal muscles that results in energy expenditure” “Exercise is a subset of physical activity that is planned, structured, and repetitive and has as a final or an immediate objective the improvement of maintenance of physical fitness. Including the distinction will make it clearer to understand. Insert Keating et al. (ref. 1) here, as this sentence is almost verbatim from the article.

Line 94 – 97: This is confusing. What timepoint is the end of the 2018 intervention? What is the purpose of Time 0, when was time 0 assessed?  Even when consulting the protocol outline figure, it is unclear what exactly the time periods where. Please be specific. Specific aim 1: assessing the 3-year effects of ‘Living Better’ seem to hinge on the end of the 1st intervention in 2018 (non-declared time -1?) and the baseline of the 2nd intervention. The purpose of time 0 remains unclear.

Page 4: “Enrollment” – Spelling Error. “Analysis” is cut off in your chart. “Analyzed”  - Spelling error.

From your flowchart ~25% of the 1st intervention cohort declined to participate in a 2nd intervention. ~11% had declined in their health at time of follow-up. This is telling that the effects of your 1st intervention were possibly not withstanding time / participants falling back into old habits once intervention concluded. Introduction of bias very likely – caution on interpretation of findings.

Line 144 – 145: What was the BP measurement device? Was it the same across all pharmacies? Are the blood pressure devices comparable in their accuracy?

Line 159: How was weight measurement standardized? Did the pharmacy technicians follow an SOP? Was clothing standardized? When were weights being taken?

Line 257 – 259: Where are the data from pre 1st intervention to back this up? It should be illustrated how pre 1st intervention baseline compares to 1st intervention end, and to 2nd intervention baseline. It would be helpful for clarity to include the data from the previous publication for the 29 participants.

Round 2

Reviewer 1 Report

1. Please rephrase Uncompleted assessments (n=7) into incomplete assessments

I do not have any further comments.

Author Response

Thank you very much for your comments. Please see below the answers.

Point 1: Please rephrase Uncompleted assessments (n=7) into incomplete assessments.

Response 1: Thank you for the comment. Effectively, as you suggested, we have rephrased that information in the Figure 2 (flowchart).